

# Data assimilation schemes for ocean forecasting: state of the art

Matthew J. Martin[1], Ibrahim Hoteit[2], Laurent Bertino[3], Andrew M. Moore[4]

[1]Met Office, Exeter, UK
[2]Physical Science and Engineering Division, King Abdullah University of Science and Technology (KAUST), Thuwal, Saudi
Arabia
[3]Nansen Environmental and Remote Sensing Center, Bergen, Norway
[4]Ocean Sciences Department, University of California Santa Cruz, Santa Cruz, California, USA

*Correspondence to*: Matthew Martin (matthew.martin@metoffice.gov.uk)

**Abstract.** Data assimilation is a process for integrating models and observations into comprehensive and reliable estimates of the ocean state. It is used to produce near-real time initial conditions (analyses) from which ocean forecasts are produced and to generate reconstructions of the past state of the ocean (reanalyses). Here we provide an overview of the methods currently
used in ocean systems for assimilating satellite and in-situ observations, together with a brief review of methods being developed which will be implemented in future operational systems, including the use of machine-learning techniques that provide a way to improve their efficiency. A list of data assimilation software used by most of the global and regional operational ocean forecasting systems is provided, together with its availability. A discussion of practical considerations for employing data assimilation software and techniques operationally is also given, including the types of observations which are
commonly used, and the implementation choices made by existing operational systems at global and regional scales is summarized.

## 1 Introduction

Accurate estimates of the state of the ocean are required for many purposes. Observations provide direct information about the ocean but are sparse in time and space. Numerical models can give information everywhere and describe the time evolution of
the ocean but are prone to error. Data assimilation (DA) is the process by which these two sources of imperfect information are combined, taking into account their errors, in order to produce complete and accurate estimates of the ocean (Moore et al., 2019; Hoteit et al., 2018; Alvarez-Fanjul et al. 2022; Stammer et al., 2016; Carrassi et al., 2018). These estimates are used to produce historical reanalyses of the ocean (Storto et al., 2019; Heimbach et al., 2019) and in near real-time to initialize forecasts (Moore et al., 2019).

Data assimilation is used in global, regional and coastal ocean forecasting systems. The characteristics of the models used in each setting can be different including the resolution, processes represented and the model components. Global models are usually coupled physical ocean/sea-ice models, with a strong move at many operational centers to coupled atmosphere/ocean/sea-ice models. Regional and coastal models usually resolve more of the higher-frequency processes which



become more important in shallow seas, and often include coupled physical/biogeochemical components. The observations available for assimilation also often have different characteristics with different technologies needed to measure the ocean closer to the coast. The methods used to initialize forecasts in these different settings have to take into account the characteristics of the model and observations available so that the important processes can be constrained.

Here we provide a brief overview of data assimilation theory to put into context the various schemes used in operational ocean forecasting centers. The data assimilation software used at many of the operational centers is also described including community open source software as well as other code developed and used at some of the main institutes. An overview of the practical considerations needed to apply data assimilation effectively in an operational setting is given. We then describe the current status of data assimilation as applied in many operational ocean forecasting centers.

## 2 Data assimilation methodology

A variety of DA methods are being used or currently tested to develop Operational Ocean Forecasting Systems (OOFSs) (Moore et al., 2019). These first followed the 3D formulation of the DA problem (3DDA) in which the ocean state at a given time is estimated based only on the available observations at that time. 3DDA is often cast as a least-squares fitting problem whose solution minimizes a composite objective function involving a data-misfit term and a regularization term representing prior knowledge on the ocean state, called the background/prior and usually taken as the most recent ocean forecast. Both terms of the objective function are generally weighted by their respective observations and background error covariances, which can be also imposed following a (stochastic) Bayesian inverse formulation of the 3DDA problem under the assumption of Gaussian observations and background errors (Moore et al., 2019; Hoteit et al., 2018). When the observational operator relating the ocean state to the observations is linear, the 3DDA problem has an analytical solution, known as the Best Linear Unbiased Estimator (BLUE); when not, this operator is either linearized to compute the Optimal Interpolation (OI) solution, or the objective function is directly minimized using a gradient-based iterative optimization algorithm to compute the 3D variational DA (3DVAR) solution.

The solution of the 4D DA problem is more advanced as it is estimated based on a set of observations that is available over a given period of time (Weaver et al. 2003). It can be computed following a straightforward extension of the 3DVAR problem by formulating an objective function in which the data-misfit term constrains the ocean model prediction to the observations in time. When the ocean model is considered perfect, only the ocean state at the start of the observation period needs to be estimated. The resulting strongly constrained (by the ocean model equations) 4DVAR solution is then integrated forward with the model beyond the observations period to compute the ocean forecasts. In contrast, the weak constraint 4DVAR problem considers model errors in the ocean model, which can be then estimated as part of the objective function minimization process. Jointly estimating the ocean initial state and model errors at every time step can quickly become computationally intractable. This was elegantly addressed by moving the optimization in the observation space, which should be of much smaller dimension in this case, using the dual formulation or Representer method (Bennett, 2005). In between the strong and weak constraint



4DVAR, a large variety of different implementations exist, for instance estimating the ocean model parameters (e.g., mixing schemes) and inputs (e.g., atmospheric forcing, open boundary conditions, bathymetry, etc.) as part of the minimization process. This has been successfully demonstrated with the MIT general circulation model (MITgcm) (Forget et al., 2015) and the Regional Ocean Modeling System (ROMS) (Moore et al., 2019). In all 4DVAR methods, the computation of the objective

function gradients required for the minimization process can be efficiently implemented through the adjoint model, governed by the adjoint equations to the ocean tangent linear model. Coding and running the adjoint model can be demanding in both human effort and computational resources.

The observational and background error covariances are key in determining the 3D and 4D DA solutions. The first sets the weights of the data-misfit terms and their correlations to avoid overfitting the observations while accounting for redundant

information (Moore et al., 2019). The second constrains the DA solution by enforcing some dynamical relationships in the initial state and/or smoothness on the estimated inputs and parameters to enable a proper propagation of the observations' information into all ocean model variables (Moore et al., 2019).

The DA methods discussed so far are designed to compute a deterministic estimate of the ocean state (the maximum a posteriori of the Bayesian inversion problem), and therefore do not provide a framework to quantify the uncertainties in the ocean

forecasts, the covariance of which could be used as the background for the next DA cycle. This sets the stage to the filtering DA methods which sequentially compute the solution of the Bayesian inversion problem by considering the observations as they become available. The filtering formulation of the DA problem allows for model and observational errors and involves computing the probability distribution of the ocean state conditioned on all previous observations. This provides a recursive framework suitable for OOFSs where the model is used for forecasting the ocean state and its error statistics (forecast step),

which are then updated with the new incoming observations based on Bayes' rule (analysis step) (Hoteit et al., 2018).

The Bayesian filtering problem can be conceptually solved by the Kalman filter (KF) when the underlying dynamical and observational models are linear and their errors are Gaussian, in which case the forecast and analysis distributions are Gaussian and the analytical form of their mean (state estimate) and covariance are available. Ocean general circulation models are however nonlinear, and the discrete dimension of the underlying ocean state can be very large. This motivated the development

of a variety of simplified and extended variants of the KF for ocean DA, either by (i) linearizing the ocean dynamics and enforcing low-rank error covariance matrices (e.g., Reduced-Order Extended Kalman - ROEK - and Singular Evolutive Extended – SEEK - filters), or (ii) using the widely celebrated ensemble KF (EnKF) methods (Hoteit et al., 2018). EnKF methods use samples to compute statistical approximations of the first two moments of the ocean state forecast and analysis distributions. Given an analysis ensemble, an EnKF integrates its members, eventually with perturbed noise to account for

model errors, forward with the ocean model for forecasting, and the resulting forecast ensemble statistics are then updated with the incoming observations using the KF analysis step. The latter is referred to as stochastic when the KF analysis step is applied on each forecast ensemble member using perturbed observations, so that the analysis ensemble covariance matches that of the KF, and deterministic (e.g., ETKF, EAKF, SEIK, DEnKF) when the KF analysis step is directly applied on the mean and



covariance of the forecast ensemble, after which a deterministic resampling step is needed to resample a new analysis ensemble

(Hoteit et al., 2018).

EnKFs are generally integrated with relatively small ensembles (~100 samples) to limit their computational cost, making their sample covariances low-rank and thus necessitating localization/covariance-tapering techniques to confine the spatial range of their correlations (Hoteit et al., 2018). To further reduce the computational requirements, EnKFs are also often implemented with static ensembles, only using the ocean model to compute the forecast starting from the analysis state (ensemble OI – EnOI

- methods), or their ensembles augmented with pre-selected static members (Hybrid EnOI-EnKF methods) (Counillon et al., 2009). On the other side of the spectrum, more sophisticated filtering methods have been also proposed to move beyond the Gaussian error assumption by employing Monte-Carlo approximations of the forecast and analysis distributions, so-called Particle Filters, or through Gaussian mixture approximations which when implemented within an ensemble framework reduce to some sort of ensemble of EnKFs (van Leeuwen, 2015). These methods are however still in testing phases and are yet to be

applied in operational settings.

4DVAR and EnKFs were proven to provide viable and robust solutions for many ocean DA applications, and most ocean centers are currently developing their operational systems around these approaches. There are benefits and drawbacks in using an EnKF or a 4DVAR. EnKFs involve flow-dependent ensemble representation of the background, though rank-deficient. On the downside, the EnKF is generally only efficient for moderate model nonlinearity because of its second-order moments

approximation of the error statistics. 4DVAR, on the other hand, should better handle nonlinearities, though the optimization of its objective function can be a complex task in the presence of strongly nonlinear dynamics (Moore et al., 2019; Hoteit et al., 2018), and can be implemented with a full-rank, albeit static, background error covariance matrix. 4DVAR further requires coding and maintaining the adjoint of the observation and forecasting models, which is quite demanding. The use of automatic differentiation in distributed HPC environments, which is receiving a renaissance in the context of machine learning, may

overcome this limitation (Heimbach et al., 2005). Finally, 4DVAR does not lend itself easily to parallelization, while the important computational cost for computing the forecast ensemble can be drastically mitigated by trivial parallelization.

There have been various attempts to merge the 4DVAR and EnKF approaches in order to combine their strengths, which introduced a new family of Hybrid Ensemble-Variational (EnVAR) methods. This includes (i) considering an ensemble of DA (EnDA) methods to obtain flow-dependent error representations, or (ii) the iterative Ensemble Kalman Filters (iEnKFs) and

Smoothers (iEnKSs) which use a forecast ensemble to describe the background statistics and apply a nonlinear optimization to the 4DVAR objective function in the ensemble space (Sakov et al., 2012), and (iii) the class of 4D Ensemble Variational (4DEnVAR) methods which also performs a set of 4DVAR optimizations in the subspace spanned by the ensemble using a set of perturbed observations (Liu et al., 2012). Different 4DEnVAR versions have been proposed (Bannister, 2017), employing hybrid background covariances, adjoint model or finite differences to compute the gradients, and different types of

perturbations.

Recently, machine learning (ML) techniques have been also considered to enhance the efficiency of the DA methods, in terms of both capacity and computations (Cheng et al, 2023). ML techniques harness neural networks' (NNs) potential at



approximating highly nonlinear functions, which may enable developing computationally less demanding forecasting models (Barthélémy et al., 2022), and backward models for efficient data fitting. NNs were also proposed as end-to-end replacement

of the analysis steps (Beauchamp et al, 2023), and to parameterize and account for model errors (Farchi et al., 2021).

## 3 Data assimilation software

Data assimilation software packages come in all sizes and flavors. A first distinction needs to be made between educational packages that can be used for methodological developments and operational codes designed for high-performance computers. We will only consider the latter category in this section. A second distinction can be made between software aimed at 4DVAR

methods and those that take the EnKF as their target algorithm. These two types of software differ in their complexity and size, and therefore adopt different development strategies. There are thus several small-sized EnKF packages and a few more ambitious 4DVAR packages on the market. The latter may also include the EnKF as a small addition to their ensemble-variational toolbox. Some of the packages (DART, PDAF, JEDI) have users in other research fields beyond ocean forecasting. See Table 1 for a list of commonly used DA software in ocean prediction systems.


**Table 1: Data assimilation software packages.**

| Software name | Target algorithm(s) | Programming language | Development community | Code availability |
|---|---|---|---|---|
| JEDI | Variational DA. | C++ | JCSDA, NOAA, NASA, US Navy and Air Force, Met Office. | Open source. https://github.com/JCSDA |
| MITgcm | Variational DA. | Fortran 90. A version in Julia is under development. | ECCO consortium, GECCO, MIT, Uni. Texas | Open source. https://mitgcm.readthedocs.io/ |
| NEMOVAR | Variational DA. | Fortran 90 | CERFACS, ECMWF, Met Office, INRIA | Not open source. |



| OceanVar | Variational DA. | Fortran 90 | CMCC, CNR | Not open source. |
|---|---|---|---|---|
| ROMS | Variational DA. | Fortran 90 | ROMS community | Open source. https://www.myroms.org/ |
| DART | Ensemble DA. | Fortran 90 | NCAR | Open source. https://dart.ucar.edu |
| EnKF | Ensemble DA. | Fortran 90 | NERSC | Open source. https://github.com/nansencenter/enkf-topaz |
| EnKF-C | Ensemble DA. | C | Bureau of Meteorology | Open source. https://github.com/sakov/enkf-c |
| NEDAS | Ensemble DA | Python, parallel | NERSC | Open source. https://github.com/nansencenter/NEDAS |
| OAK | Sequential DA. | Fortran 90 | U. Liège | Open source. https://github.com/gher-uliege/OAK |
| OpenDA | Ensemble DA. | Java | TU Delft | Open source. https://www.openda.org |
| PDAF | Ensemble DA. | Fortran 90 | AWI | Open source. https://pdaf.awi.de/trac/wiki |
| SAM2 | SEEK filter. | Fortran 90 | Mercator Ocean International, ECCC. | Not open source. |
| Sequoia | Sequential DA. | Fortran 90 | OMP/LEGOS | Available on demand. https://sirocco.obs-mip.fr/ |

The above software packages have mainly been used on high-performance computers (HPCs) and some of them on personal computers. The NEMOVAR and MITgcm 4DVAR codes, and the NEDAS ensemble code, are actively being developed for
use on GPU-based systems. However, all the DA software listed above have lived long enough to be ported several times to different HPC architectures with different compilers and can be qualified as portable.





## 4 Practical implementations in operational systems

Several factors dictate the practical implementation of ocean DA systems within an operational environment. The primary controlling factors in any operational environment typically relate to (i) scheduling of the DA analysis and forecast phases
with respect to the competing demands of other essential activities (e.g. numerical weather prediction, hydrological forecasts, etc); and (ii) the release of analysis-forecast products in a timely manner so that they are of maximum benefit to the users. These overarching criteria therefore, in turn, dictate the configuration of the forecast model and the data assimilation approach that may be used.

In the case of ensemble approaches, such as the EnKF or EnVar, there may be a trade-off between model resolution and the
ensemble size in that computation time increases with resolution. Thus, with limited resources fewer ensemble members can be run within the constraints imposed by (i) and (ii). An advantage of ensemble approaches is that each ensemble member can be computed independently, meaning that in very large HPC environments, many ensemble members can be run simultaneously. Here again though there can be a trade-off between resolution and ensemble size. While most ocean models scale reasonably well on parallel computing architectures, wall-clock time typically does not scale linearly with the number of
cores. Hence, there is a point of diminishing returns whereby it may be better to allocate fewer cores to the business of computing ensemble members at the expense of a longer wall-clock time for each member, rather than dedicating a very large number of cores to a single task.

Unlike ensemble methods, the traditional approaches to variational data assimilation, namely 3DVAR and 4DVAR, are strictly sequential and cannot be parallelized in time. In other words, the inner- and outer-loop iterations of the cost function
minimization algorithm must be performed sequentially. The sequential iterative nature of variational approaches therefore imposes a heavy computational burden on the data assimilation phase of the analysis-forecast cycle, especially in the case of 4DVAR. This burden is alleviated in some 4DVAR systems by performing the inner-loop minimization steps at lower model resolution – for example, a reduction of the horizontal resolution by a factor of 2 typically yields a factor of 8 reduction in wall-clock time assuming that the inner-loop time step can be halved also. Performing the inner-loops at lower arithmetic
precision (i.e. 32 bit arithmetic versus 64 arithmetic) can lead to further cost savings. In 4DVAR, the inner-loop iterations involve integrations of the tangent linear (TL) and adjoint (AD) versions of the forecast model. Further reductions in computational cost can therefore also be achieved by reducing the complexity of the TL and AD models. Time-parallel formulations of 4DVAR based on a saddle-point algorithm also yield substantial computational savings (Fisher and Gurol, 2017; Moore et al, 2023).

The assimilation strategy employed also depends on the type of observations that are to be assimilated, and their distribution in time. In the case of a Kalman filter, while each observation can be assimilated sequentially at the associated observation time, this may not be an efficient strategy since this might require overly frequent stopping and restarting of the filter computations. Thus, it is often preferable to group together observations that are closely spaced in time and treat them as though they were available at the same time. This approach underpins the strategy of First-Guess at Appropriate Time (FGAT)





which is commonly employed in conjunction with both ensemble approaches and 3DVAR. Such approaches necessitate the choice of a time-window over which the observations will be aggregated for assimilation. In between times, the forecast model is run to yield the first-guess or background for the next data assimilation cycle, so the time-window of aggregation also dictates how frequently the analysis-forecast cycle can be performed. For an EnKF, it is sufficient to store observation equivalents from each model ensemble member to calculate asynchronous cross-covariances (Sakov et al. 2010). In the case

of 4DVAR, observations are typically assimilated at the actual time of observation. This involves integrations of the TL and AD models forward and backward in time. Since these are based on a linearized version of the forecast model, the validity of the linear assumption through time is an important consideration. In particular, linear instabilities can develop if appropriate care is not exercised. Therefore, while a long-time window in 4DVAR may be preferable so that the analysis is informed by more observations, this must be balanced by the validity of the linear assumptions employed in the TL and AD models, and

the added computational burden of the longer assimilation window.

**5 Ocean Observations**

While there is a common subset of observations from the global ocean observing system (GOOS) that are assimilated into ocean models, additional sources of data may be available for assimilation into regional ocean models that are not appropriate for global models. The GOOS and different types of observations available are discussed in the ETOOFS guide (Alvarez-

Fanjul et al., 2022). The mainstay of the GOOS is remote sensing observations of sea surface temperature (SST), sea surface height (SSH), sea surface salinity (SSS) and sea ice concentration. This is supported by the Argo network of profiling floats that provide vertical sections of temperature and salinity (and in some cases biogeochemical variables) mostly over the upper 2000 m of the water column, although deep Argo floats below 2000 m are now also being deployed. In the tropical oceans, the observing system is augmented by networks of buoys that provide profiles of temperature (and in some cases salinity and

currents) to depths of ~500 m. Observations from tagged marine mammals also provide useful information in some regions of the world ocean. In coastal regions, other data sources are often available that cannot be readily assimilated into global models because of the disparity in horizontal resolution. These include data from gliders and other autonomous underwater vehicles (AUVs), estimates of surface currents from high-frequency (HF) radars, other tagged marine mammals, moorings, drifters, and in some locations dedicated coastal arrays.

All observations, regardless of their origin, must be subject to strict quality control (QC) standards before they can be assimilated into a model (Good et al., 2023). All operational centers employ sophisticated QC systems for flagging and rejecting erroneous observations and those of poor quality. In addition, the large volume of remote sensing observations from earth orbiting satellites must generally be thinned in space and time. There are three main reasons for this: first, remote sensing observations contain a great deal of redundancy which can be reduced by judicious thinning; second, the sheer volume of

remote sensing observations can quickly overwhelm a data assimilation system if not appropriately thinned (particularly in light of the high redundancy); and lastly, accounting for correlated observation errors in data assimilation systems is technically



challenging, so thinning the observations is one approach for reducing the degree of correlation. Another important aspect of operational data assimilation systems is the formation of so-called "super observations". This refers to the procedure for combining multiple observations of the same type that fall within a model grid cell at the same observation time into a single
datum (a super observation). This usually entails some simple averaging or aggregation procedure and is necessary in order to improve the numerical conditioning of the data assimilation inverse problem.

Since the observations are the only, albeit far from complete, measure of the true state of the ocean, they often form the basis for metrics that are used to monitor the performance of data assimilation systems. The statistics of the observation minus background (OmB) and observation minus analysis (OmA) provide information about the fit of the model to observations
before and after the observations have been assimilated. The statistics of OmB and OmA provide an important diagnostic check on prior assumptions made about the background error and observation error covariances. Inconsistencies between the actual and expected error statistics can be used to retune the data assimilation system, regardless of the data assimilation methodology employed. In variational data assimilation systems, continuous monitoring of the cost function and cost function gradient also provide useful diagnostics of system performance. The impact of different components of the observing system
can also be quantified and monitored in various ways. This is commonly done in terms of the impact on the skill of forecasts that are initialized from the data assimilation analyses. By continuously monitoring the impact of each component of the observing system on forecast skill, data streams that consistently degrade the forecast skill can be flagged (and removed), and the degradation of any data stream over time can be identified.

## 6 Current status of data assimilation in operational forecasting systems

An overview of operational ocean data assimilation systems and their characteristics is provided in Figure 1 for global systems and Figure 2 for regional and coastal systems. Not all operational systems are covered here, but the figures provide information about the main choices which have been made by existing operational centers producing near-real time forecasts in the configuration of their data assimilation schemes. The information represents the current operational status, but all centers are continually developing and improving their systems, and many have research configurations which are more sophisticated
than those presented.

In general, the global systems use somewhat simpler DA algorithms (though they are still complex in their implementation of those algorithms) than the regional and coastal systems. Many global forecasting groups use a 3DVAR-FGAT algorithm (Barbosa Aguiar et al., 2024; Zuo et al., 2019; Cummings and Smedstad, 2013; Storto et al., 2016; Ravichandran et al., 2013) with some groups using a SEEK filter or LESTKF with a static ensemble (Lellouche et al., 2018; Smith et al., 2016; Li et al.,
2021) or EnOI schemes (Chamberlain et al., 2021). The reason these algorithms are simpler is largely due to the large number of grid points, especially in the higher resolution global systems, which restricts the options for more expensive algorithms when timely delivery of forecasts is the main goal. Some groups are testing more sophisticated schemes in research mode though, including those which make use of ensembles, e.g. MOI are testing LETKF, Met Office and ECMWF are testing



hybrid-3DEnVAR schemes (Lea et al., 2022), BoM are testing use of EnKF, and JMA are implementing 4DVAR (Fujii et al.,
2023). The observations assimilated in these systems are fairly consistent across the different systems, with the main difference
being whether the systems include sea-ice or atmosphere components. Some of the DA systems are focussed purely on the
ocean, many include a sea-ice component, and some now run with a coupled atmospheric component, though these systems
all still use so-called "weakly" coupled DA where the DA in the atmospheric and ocean/sea-ice components are run separately,
despite using coupled models (see for example Guiavarch et al., 2019 and de Rosnay et al., 2022). There is a large range of
time windows used by the different systems, with the most common time window being 1 day, a short 6-hour window used in
the Met Office coupled DA system (to match the time window in the atmospheric DA; Guiavarc'h et al., 2019), and with longer
time windows of 5-7 days used by some systems.

There is a wider range of DA algorithms employed in regional and coastal forecasting systems from EnOI/static SEEK filters
(Carvalho et al., 2019; Ji et al., 2017; Smith et al., 2021; Escudier et al., 2022) and 3DVAR-FGAT schemes (Rahaman et al.,
2018; King and Martin, 2021; Coppini et al., 2023) through to the more sophisticated EnKF (Röhrs et al., 2023) and 4DVAR
algorithms (Moore et al., 2023; Iversen et al., 2023; Hirose et al., 2019; Lee et al., 2018). Many of these regional systems also
include biogeochemical DA (see Fennel et al., 2022 for a recent review), and some include coupled sea-ice DA (e.g. Bertino
and Xie, 2020). The range of observations assimilated is also quite varied with some systems only assimilating SST data while
others include the full range of available observations including HF radar, gliders and biogeochemical data from satellites and
in situ platforms.





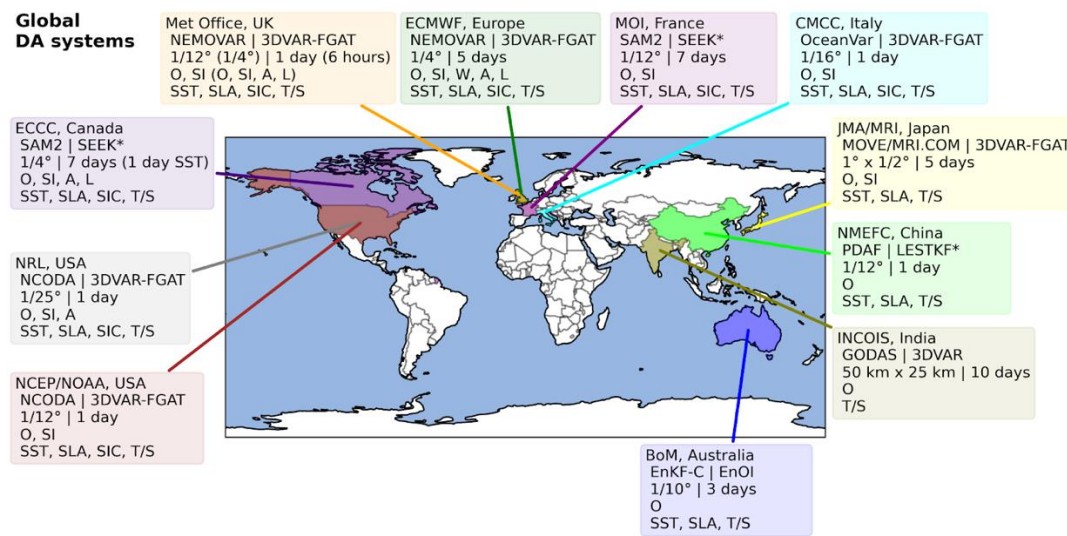


**Figure 1: Operational global ocean data assimilation systems.** For each institute, the following are listed: the DA algorithm (* indicates fixed basis version of the algorithm) and software; DA resolution and time window; Earth system components (O = physical ocean, SI = sea-ice, A = atmosphere, W = surface waves, BGC = ocean biogeochemistry, L=land); observations assimilated (SST = sea surface temperature, SLA = sea level anomaly, SIC = sea-ice concentration, SID = sea-ice drift, T/S = profiles of temperature and salinity, OC = satellite ocean color, BGC = biogeochemical profile data, HFR = HF radar).

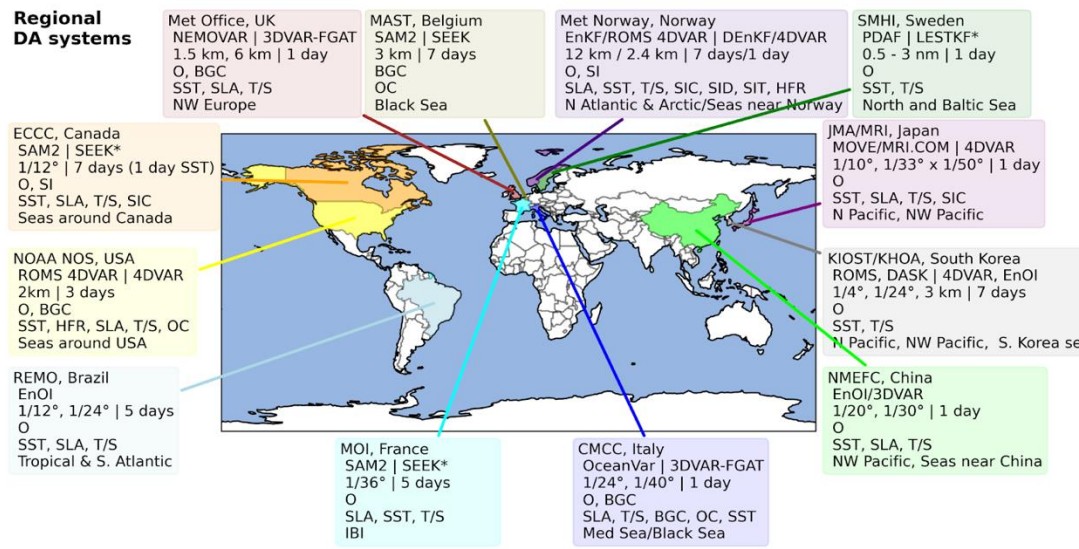

**Figure 2: Operational regional and coastal ocean data assimilation systems. See description for Figure 1.**



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

**Competing interests**

The contact author has declared that none of the authors has any competing interests.

**Data and/or code availability**

No data or code were used to produce the manuscript, but a list of data assimilation software is provided in Table 1, together with its availability.

**Authors contribution**

All authors contributed to the writing and review of the manuscript. MM organised the manuscript and led the writing of sections 1 and 6. IH led the writing of section 2. LB led the writing of section 3. AM led the writing of sections 4 and 5.

**Acknowledgements**

The authors would like to thank the compilation team of the OceanPrediction Decade Collaboration Centre who organised this special issue.