# Peer review of "Data assimilation schemes for ocean forecasting: state of the art"

_State of the Planet, 2024_

## Author Response (AR1)

**Response to review RC1.**

We thank the reviewer for the useful comments on the paper. The comments are copied in black below and our responses are in blue.

**General comments:**

The international landscape of operational oceanographic forecasting systems (and the data assimilation schemes used) has evolved considerably over the last ten years. It is thus timely that Martin et al. provide this overview and description of the different methods in place and how they are implemented globally and regionally. The review provides a thoughtful and accurate description of the various methods applied, together with comments regarding particular challenges and benefits of each. Overall, I find the paper provides a useful reference regarding the current state of ocean data assimilation systems and their implementations. Below I've provided a few comments the authors may want to consider to further improve the breadth and impact of the paper.

Thanks for these comments and for the suggestions to improve the paper.

**Main points**

1. It is not clear to me reading this paper to what readership it is intended. The introduction provides a quite high-level background on operational oceanography. This is then followed by a description of methods, which is often quite technical (e.g. reference to model adjoints, etc..). It ends without any kind of summary, discussion or perspective toward the future. It reads as though it is a chapter in book, but I didn't see any reference to a Special Issue. I'm not familiar with this journal and it appears to organize papers into chapters somehow. Regardless, it would be helpful that the paper is either more self-contained or references the contextual pieces.

The article is part of a Report (a special issue) titled "Ocean prediction: present status and state of the art". This special issue is introduced in a paper by Alvarez Fanjul and Bahurel (2024). We have now included a reference to that introductory paper in our introduction and have included a new "Future directions" section at the end with a perspective towards the future.

- 2. The introduction is quite general while the rest of the paper digs into some fairly detailed comments about the methods. There is little, however, to provide context regarding the particular DA challenges for ocean prediction and its history. Rather the paper focuses almost exclusively on the methods themselves.
  - 1. For example, ocean DA methods have evolved in large part from those developed for Numerical Weather Prediction. Some mention to this effect would be appropriate.
  - 2. The impact of particular challenges of ocean DA on the methods used would be relevant. For example, the spatial and temporal scales of the ocean are quite

different than for the atmosphere. The observations available are also quite different with satellite observations only providing surface information. This affects both the cost / benefit of using ensembles as compared to higher model resolution as well as limitations on constrained scales due to information content of satellite altimetry. While I understand the focus of the paper is on DA methods, the choice of method depends on the model used and what observations are available. Some discussion on this point could be made in Section 5 on Ocean Observations as well.

We have included a new paragraph in the introduction to give the context as suggested by the reviewer:

"Many of the data assimilation methods used in ocean forecasting were originally developed for numerical weather prediction (with the notable exception of the ensemble Kalman Filter). The dominant spatial and temporal scales in the ocean are quite different to the atmosphere though, with the first baroclinic Rossby radius of deformation being a few 10s of kilometers at midlatitudes (see e.g. Chelton et al., 1998) with temporal scales ranging from days to weeks. To resolve the open-ocean mesoscale at mid-latitudes, model resolutions of the order of at least 1/12° are required (see e.g. Hewitt et al., 2016) and the aim of many global ocean data assimilation systems is to initialise the ocean state at these scales. Observations of the surface ocean are available at fairly high resolution from satellites, but observations of the most of the observations to constrain models of the 3D ocean on the desired scales. The integration of high-resolution models along with the high computational processing required for implementing an advanced data assimilation method demands computational resources that are available at only a small number of ocean forecasting centers and research institutions worldwide "

3. Another area I felt was underrepresented was a discussion of errors. DA aims to provide a best estimate of the system state based on estimates of model and observational errors. In this sense, it could be helpful to outline what the key errors are that are being accounted for (e.g. due to model physics, resolution, forcing, intrinsic variability). Some comment regarding the treatment of bias and representivity error would also be relevant.

We have included a new paragraph in the introduction to discuss the various sources of error as suggested by the reviewer:

"Errors in ocean models arise due to approximations in their numerical formulation, errors in the parameterisation of unresolved physics, and errors in the inputs to the model including the surface atmospheric forcing, river inputs, and the lateral boundary conditions for regional systems. The ocean is a chaotic system, so small differences in the initial state grow over time, especially in strongly eddying regions. All these sources of uncertainty contribute to the model forecast error, estimates of which are needed for data assimilation. Observations also contain errors and measure the ocean on different spatial scales (to each other and to the model). Estimates of the errors in the different observations are therefore also needed, including the component due to the measurement itself, as well as the component due to the difference in the representation of the ocean by the observation and model (Janjić et al., 2018). "

We have also included a paragraph in the Observations section discussing the observation errors (including the representation errors).

4. I found the lack of any kind of summary, conclusions or discussion section to be quite unusual. Perhaps some historical context to note how quickly the landscape of operational oceanographic systems has been changing and how developments in satellite observing systems (e.g. SWOT, surface current retrievals) and ML methods may lead to further rapid developments. Challenges associated with constraints on the submesoscale circulation (which affects many applications) could also be mentioned as a limitation of current ocean DA.

We have included a "Future directions" section as suggested. This gives a brief summary of some of the areas of ongoing development including resolution increases, ensembles, new observation types and the use of machine learning.

**Minor comments**

Intro line 37 "important processes can be constrained." It is not really a process that is constrained but rather the associated variability in essential ocean variables.

Changed to "the variability associated with the important processes can be constrained."

Line 59: "When the ocean model is considered perfect...", I would specify "When the ocean model AND ATMOSPHERIC FORCING is considered perfect..."

Changed to "the ocean model and its forcing are considered perfect"

Line 70: "...adjoint model, governed by the adjoint equations to the ocean tangent linear model". These terms are introduced without any explanation. Some background or at least a reference would help readers to follow.

We've included two references for ocean adjoint and tangent linear models (Moore et al., 2004; Vidard et al., 2015).

**Response to review RC2.**

We thank the reviewer for the useful comments on the paper. The comments are copied in black below and our responses are in blue.

**General comments**

The paper provides a comprehensive review of the current state of data assimilation techniques used in operational ocean forecasting centers. It highlights the main methods in the field and includes a section dedicated to the role of observation systems. It also discusses the practical implementation of data assimilation methods in operational systems. The text is generally clear and easy to read. Nevertheless, it is recommended that the authors carefully review both the language and the references. Overall, the manuscript offers a valuable and relevant review for the community working with data assimilation methods in operational oceanography. The paper requires MINOR REVISIONS and will be ready for publication once the following points and comments are considered.

Thanks for these comments and for the suggestions to improve the paper.

**Specific comments**

In the "Data assimilation methodology" section, the discussion is well done. Nevertheless, the authors can consider including a table listing the main advantages and disadvantages of the primary methods: 4D-Var and EnKF. Alternatively, if they prefer to leave the text as it is, they should at least reference previous works that have made such comparisons as an example https://doi.org/10.1111/j.1600-0870.2007.00261.x (Kalnay et al., 2007).

We have included references discussing advantages/disadvantages of 4DVar and the EnKF (Lorenc, 2003; Kalnay et al., 2007).

Section 2 does not mention the applicability of B-matrix inflation in ensemble-based methods, which is often necessary due to the limited ensemble size or numerical corrections.

We have included a sentence on the need for ensemble inflation:

"Limited ensemble size can also result in underestimation of the ensemble variance leading to the need for ensemble inflation (Evensen et al., 2022)."

The authors raise several points on the pratical implementations, such as the trade-off between ensemble size and model resolution. However, it is important that they emphasize the information in the following paragraph earlier, perhaps in the introduction or at the beginning of the section 4.

Ocean variability is primarily driven by mesoscale eddies, with spatial scales on the order of 50-200 km and temporal scales ranging from days to weeks. Therefore, the integration of a highresolution OGCM (Ocean General Circulation Model), capable of accurately reproducing ocean variability, along with the high computational and mathematical processing required for implementing an advanced data assimilation method, demands computational resources that are available at only a few ocean forecasting centers and research institutions worldwide.

Thank you for these suggestions. We have included a paragraph in the introduction (also in response to a similar comment by reviewer RC1) to capture these points:

"Many of the data assimilation methods used in ocean forecasting were originally developed for numerical weather prediction (with the notable exception of the ensemble Kalman Filter). The dominant spatial and temporal scales in the ocean are quite different to the atmosphere though, with the first baroclinic Rossby radius of deformation being a few 10s of kilometers at midlatitudes (see e.g. Chelton et al., 1998) with temporal scales ranging from days to weeks. To resolve the open-ocean mesoscale at mid-latitudes, model resolutions of the order of at least 1/12° are required (see e.g. Hewitt et al., 2016) and the aim of many global ocean data assimilation systems is to initialise the ocean state at these scales. Observations of the sub-surface ocean are available at fairly high resolution from satellites, but observations of the most of the observations to constrain models of the 3D ocean on the desired scales. The integration of high-resolution models along with the high computational processing required for implementing an advanced data assimilation method demands computational resources that are available at only a small number of ocean forecasting centers and research institutions worldwide."

For the "Ocean observations" section, it is important to address the components of observation errors, including instrumental, representativeness (for instance, after the discussion on superobs lines 218-220), and time-dependent errors. Are there any recent recommendations or updates in the literature regarding the definition of instrument error for each variable/platform? Provide typical values or refer to recommended values in the literature.

Following a similar comment by reviewer RC1 we have included a paragraph in the introduction which discusses the sources of errors in the model forecast and the observations, including the instrumental and representation components of the observation errors. We have also now included a paragraph in the observations section with more detail:

"The use of observations in data assimilation requires information about their uncertainties. The observation uncertainty consists of a component due to the instrument error and a component related to the different representation of the ocean by the observations and the model (for example representing different spatial and/or time scales; Janjić et al., 2018). Some observation types (e.g. satellite SST) are provided together with information about the expected uncertainty in each measurement and this information can be used directly in the data assimilation. For other observation types, estimates of the uncertainty have to be obtained from the literature. An example list of instrumental uncertainties for different observation types assimilated in a global ocean forecasting system is provided in Table 1 of Lea et al. (2022)."

Section 5 does not mention sea level anomaly data from SWOT, which will be a crucial source of observations for future ocean forecasting. Additionally, data assimilation systems typically

assume a diagonal observation covariance matrix, which is not suitable for the assimilation of SWOT data due to its spatial correlations. Incorporate this discussion briefly into the text.

We have included a new "Future directions" section at the end which includes mention of SWOT as well as the representation of spatially correlated observation errors in data assimilation.

**Community comments**

CC1 by Pavel Sakov.

This is a quick comment regarding the top figure on p. 11 (l. 275).

On 29 June 2022 Australian Bureau of Meteorology has transitioned its operational global ocean forecasting system from OceanMAPS v3.4 to OceanMAPS v4.0. OceanMAPS v4.0 is a hybrid EnKF/EnOI system with 48 dynamic members and 144 static members. OceanMAPS v4.0 is performing beautifully since then and is about to be upgraded to v4.1 with a 1-day cycle. Therefore, the entry for Australia on the above figure needs to be changed to "BoM, Australia. EnKF-C | EnKF. 1/10 degree | 3 days. SST, SLA, T/S."

Thanks for the correction on this. I've updated the figure to replace EnOI with EnKF and adjusted the text starting line 242 (of the original manuscript) to read:

"In general, the global systems use somewhat simpler DA algorithms (though they are still complex in their implementation of those algorithms) than the regional and coastal systems, the exception being the BoM system which uses a hybrid-EnKF with 48 dynamic members and 144 stationary low-mode members (Brassington et al., 2023). Many global forecasting groups use a 3DVAR-FGAT algorithm (Barbosa Aguiar et al., 2024; Zuo et al., 2019; Cummings and Smedstad, 2013; Storto et al., 2016; Ravichandran et al., 2013) with some groups using a SEEK filter or LESTKF with a static ensemble (Lellouche et al., 2018; Smith et al., 2016; Li et al., 2021)."

And we've included the reference:

Brassington, G. B., Sakov, P., Divakaran, P., Aijaz, S., Sweeney-Van Kinderen, J., Huang, X., and Allen, S.: OceanMAPS v4. 0i: a global eddy resolving EnKF ocean forecasting system, in: OCEANS 2023-Limerick, IEEE, 1–8, https://doi.org/10.1109/OCEANSLimerick52467.2023.10244383, 2023.

**CC2 by Lars Nerger.**

Looking through the manuscript I found a few aspects I like to comment about:

1. line 91 mentions 'Reduced-order extended Kalman filter (ROEK)'. Actually, this very early development never seemed to attain any relevance. As such it's surprising to find this in a manuscript intending to become a reference paper. Actually, the concept of error-subspace DA (linked to SEEK) seems to be more relevant (we also find this e.g. in the publications about

'unstable subspace'). Considering the error-subspace SEEK and EnKFs are also rather similar as both represent an error-subspace (but the linearized dynamics used in the SEEK filter tend to yield worse performance as was by demonstrated by Nerger et al. (2005). A Comparison of Error Subspace Kalman Filters, Tellus A, 57A(5), 715-735, doi:10.1111/j.1600-0870.2005.00141.x)

We have removed the mention of the ROEK.

2. In line 92 the manuscript cites 'Hoteit et al., 2018' for the EnKF. This is a book chapter that was not peer reviewed. Obviously there are relevant original peer-reviewed articles (e.g. Evensen, 1994, Burgers et al, 1998, Houtekamer & Mitchell, 1998) about the EnKF, but also peer reviewed articles that provide an overview of the state of the art of EnKFs (e.g. Vetra-Carvalho et al., Tellus A, 2018, doi:10.1080/16000870.2018.1445364). Scientific diligence would perhaps rather call for citing such peer-reviewed references.

We have changed the citation to Vetra-Carvalho et al., 2018.

3. Regarding particle filters (~line 108), van Leeuwen (2015) is cited. Actually, since then there was progress in the developments. Van Leeuwen et al. (2019) Particle filters for high-dimensional geoscience applications: a review. Quarterly Journal of the Royal Meteorological Society, 145, 2335-2365, doi:10.1002/qj.3551 gives a more comprehensive review including examples from high-dimensional applications of the PF (in the atmosphere, not the ocean).

We have changed the citation to Van Leeuwen et al., 2019.

4. The description of PDAF in Table 1 is not fully correct: PDAF uses features of Fortran 2003. Further, PDAF targets not only ensemble DA, but also includes 3D-Var. For the link to the website, I recommend to only state 'https://pdaf.awi.de', the sub-directory '/trac/wiki' is an automatic forwarding (and while the domain will certainly be conserved the sub-directory might disappear in case that we update the software running on the server). It would be good if these aspects could be taken into account.

You raise a good point that we have specified Fortran 90 in the table while many of those codes (including PDAF) make use of more modern features. To address this we have removed the specific Fortran standard from all the relevant table entries.

We have updated the link to 'https://pdaf.awi.de'.

5. Figure 2 on operational regional and coastal ocean DA systems seems to be rather incomplete and it's not clear why the particular systems were chosen. E.g. in Germany a regional system for the North Sea and Baltic Sea is run (e.g. described in Bruening et al., Hydrographische Nachrichten 118 (2021) 6-15, doi:10.23784/HN118-01) This system uses a fully dynamic LESTKF with PDAF. Also next, to the SMHI-System that is mentioned, there is the operational system for the Baltic Sea of the EU Copernicus program run by the Baltic Monitoring and Forecasting Center (BAL-MFC, see e.g. https://marine.copernicus.eu/; there doesn't seem to be a proper peer-reviewed article, but the center descriptions and the documentation of the data products), but also the Danish Meteorological Institute seems to run their own system. Apart form this NMEFC

also runs an ensmeble-based system for sea ice in the Arctic ocean (e.g. Liang, X., Z. Tian, F. Zhao, M. Li, N. Liu, C. Li (2024) Evaluation of the ArcIOPS sea ice forecasts during 2021–2023. Front. Earth Sci. 12, 1477626 doi:10.3389/feart.2024.1477626). Actually, all these systems use PDAF leveraging commonalities

As stated in the original manuscript, we have not attempted to provide a complete list of all operational systems which is particularly difficult for the regional and coastal systems. We have now made this clearer in the text by adjusting the following sentence:

"Not all operational systems are covered here, but the figures provide information about the main choices which have been made by some of the existing operational centers producing near-real time forecasts in the configuration of their data assimilation schemes."

We have also now included the BSH system in the figure, together with the Bruening et al. reference suggested above.